# Comprehensive Investigation of Parameters Influencing Fluorescence Lifetime Imaging Microscopy in Frequency- and Time-Domain Illustrated by Phasor Plot Analysis

**DOI:** 10.3390/ijms232415885

**Published:** 2022-12-14

**Authors:** Thomas Kellerer, Janko Janusch, Christian Freymüller, Adrian Rühm, Ronald Sroka, Thomas Hellerer

**Affiliations:** 1Multiphoton Imaging Lab, Munich University of Applied Sciences, 80335 Munich, Germany; 2Faculty of Physics, Soft Condensed Matter, Ludwig-Maximilians-University, 80539 Munich, Germany; 3Laser-Forschungslabor, LIFE Center, Department of Urology, University Hospital, Ludwig-Maximilians-University, 82152 Planegg, Germany; 4Department of Urology, University Hospital, Ludwig-Maximilians-University, 81377 Munich, Germany

**Keywords:** fluorescence lifetime imaging microscopy, FLIM, fluorescence microscopy, two-photon microscopy, phasor-plot, image analysis, time-domain, frequency-domain, bioimaging

## Abstract

Having access to fluorescence lifetime, researchers can reveal in-depth details about the microenvironment as well as the physico-chemical state of the molecule under investigation. However, the high number of influencing factors might be an explanation for the strongly deviating values of fluorescent lifetimes for the same fluorophore reported in the literature. This could be the reason for the impression that inconsistent results are obtained depending on which detection and excitation scheme is used. To clarify this controversy, the two most common techniques for measuring fluorescence lifetimes in the time-domain and in the frequency-domain were implemented in one single microscopy setup and applied to a variety of fluorophores under different environmental conditions such as pH-value, temperature, solvent polarity, etc., along with distinct state forms that depend, for example, on the concentration. From a vast amount of measurement results, both setup- and sample-dependent parameters were extracted and represented using a single display form, the phasor-plot. The measurements showed consistent results between the two techniques and revealed which of the tested parameters has the strongest influence on the fluorescence lifetime. In addition, quantitative guidance as to which technique is most suitable for which research task and how to perform the experiment properly to obtain consistent fluorescence lifetimes is discussed.

## 1. Introduction

Modern microscopy excels at uncovering the morphology of living samples [1]. Today, advanced techniques, such as stimulated Raman scattering (SRS) [2], fluorescence correlation spectroscopy (FCS) [3,4] and fluorescence lifetime imaging microscopy (FLIM) [5], give additional access to photo-physical, chemical and biological information, to mention just three types [6]. Furthermore, this information can be quantified, e.g., by measuring the local concentration of a molecular species in an area as small as one femtoliter [7]. Unfortunately, the majority of microscopy methods rely on signal intensities, which are dependent on the experimental setup, such as the excitation/detection efficiencies, or susceptible to unwanted phenomena, such as photo-bleaching of the markers used [8]. Therefore, only relative changes are commonly quantified. FLIM is an exception here because the measured lifetime is not dependent on the kind of excitation, e.g., via one-photon (1P) or two-photon (2P) absorption or on matching the laser wavelength to the excitation profile of the sample. Furthermore, different kinds of detection, e.g., single photon counting with a point-detector or a camera-based widefield approach, deliver the same results. This paper refers to this desirable feature as the FLIM-advantage. Why then is the lifetime of a specific fluorophore not the same under all circumstances? Consulting the literature, one finds differing values that lead to the impression that FLIM may not be a reliable method. As a benchmark, two examples are represented: the reported lifetime of rhodamine B ranges from 1.74 to 3.13 ns [9,10,11,12], or the lifetime of rose bengal ranges from 0.095 to 2.4 ns [13,14,15,16]. Despite these huge differences, there is no evidence that the authors of the published data made any mistake in their measurements or conclusions. Therefore, the aim of this paper is to clarify how these discrepancies arise and what circumstances lead to differing lifetime values. On the other hand, it will be shown that time domain (TD) and frequency domain (FD) FLIM lead to the same results although they employ the different setup parameters mentioned above. Unfortunately, they are not directly comparable because the corresponding data analysis is different. Therefore, the data of both methods are transformed to retrieve a so-called phasor plot, which makes the comparison an easy task. The reason for the superficial discrepancy in lifetimes is due to certain parameters that influence the sample directly, such as concentration, pH-value, solvent polarity, and temperature, etc. This fact will be demonstrated in our comprehensive investigation. On the other hand, if these parameters are under the control of the researcher, they can use this to their advantage and utilize the fluorophore as the smallest possible reporter inside a living cell.

## 2. Results

All parameters that were studied in this rigorous investigation fall within two categories—setup-dependent and sample-dependent. The former determines the accuracy and range in which the lifetime can be obtained. The latter includes all parameters related to the microenvironment and the photo-physical properties of the fluorophore that affect its lifetime.

### 2.1. Setup Dependent Parameters

Starting with the different measurement setups, interdependent advantages and disadvantages of each were worked out. While the TD setup stands out with its diffraction-limited resolution, deep tissue penetration and optical sectioning possibilities, the needed detection time must be long enough to gather sufficient photon statistics. If a high resolution is not needed, the fast image acquisition of the FD setups provides the greatest advantage.

#### 2.1.1. Wavelength

An often-formulated question is whether the kind of absorption process, e.g., 1P- or 2P-absorption, or the excitation wavelength, affect the fluorescence lifetime [12]. Therefore, the setups chosen in this series differ in the way the excitation takes place. For the TD setup, two ultrashort pulsed lasers with center wavelengths at 780 and 1034 nm were utilized, whereas the FD setup employed two wavelengths at 405 and 445 nm. To compare the lifetimes resulting from two-photon (TD FLIM) and one-photon (FD FLIM) absorption, a single dye with suitable spectral properties was used (Figure 1A). Rose bengal dissolved in ethanol at a concentration of $10^{-2}$ M met this requirement. The lifetimes determined using all four wavelengths, as well as one-photon and two-photon absorption, gave consistent results of around 0.85 ns (Figure 1B). A *p*-value test yielded a result of 0.046, which met the requirement for the measurement to be considered statistically significant. To also show the mono-exponential behavior within the FD, a *p*-value between the phase and demodulation lifetime was determined (Table 1). The mono-exponential behavior in the TD was verified by the residuals of the single exponential fit.

#### 2.1.2. Modulation Frequency

The modulation frequency, with which the sample is excited, plays an important role; hence, it defines in which range the fluorescence lifetime can be detected. For the TD, the limiting factor is the repetition rate of the laser source. All photons detected during an excitation cycle can be accurately assigned to a specific excitation-pulse and thus counted for the statistics. If the time between pulses increases (i.e., the repetition rate decreases), longer lifetimes can be measured and vice versa.

Because of the tangent function in Equation (Equation 17), it can be derived that a smaller modulation frequency must be selected for longer lifetimes. If this criterion is not met, the measurement results are more scattered. The reason for this could be explained easily by the phase lifetime. While the frequency increases, the working line becomes steeper due to the stretched tangent function. The phase determination is, therefore, less accurate and results in scattered lifetimes with bigger errors, as can be seen in Figure 1C and Table 1.

For statistical evaluation, the *p*-value analysis for all phase- and demodulation-lifetimes was performed. The overall *p*-value (Table 1, last column) is the statistical proof that, for the different frequencies, the same lifetime is detected, while the single *p*-value (Table 1 bottom row) represents the consistent results between the phase- and demodulation-lifetimes. Due to the obviously too-large standard deviation for the frequency of 1 MHz, the corresponding raw data were left out for the calculation. This resulted in an overall *p*-value of 0.048 for the phase lifetime and 0.046 for the demodulation lifetime in a frequency range of 10 to 40 MHz. For the TD, a fluorescence lifetime of 0.633 ns was obtained for a frequency of 80 MHz. This value coincides with those from the FD.

The experimental results imply that the lifetime is not dependent on the modulation frequency as long as the repetition rate of the pulsed laser in TD or the modulation frequency of the laser in FD are adequately chosen.

#### 2.1.3. Integration Time

The time period for how long the fluorescence signal is collected is given by the exposure time of the camera, whereas in TD, it is the time interval over which the laser focus remains at each sample point to gather sufficient events for building the histogram. For short integration times down to 0.3 s, only FD measurements are presented, whereas for longer times, TD measurements only up to 30 s are shown. Here, the expectation would be that the longer the integration time, the more accurate the determination of the lifetime due to increased statistics. This is confirmed for TD but not in the same manner for FD, as can be clearly seen in Figure 1D, with increasing error bars for FD measurements. Here, a saturation effect caused by the limited dynamic range of the camera plays an important role, which will be discussed later in detail.

Again, the *p*-value is used to make statistical statements. Here, the raw data at 300 ms exposure time in the FD were not taken into account due to the high standard deviation caused by pixel saturation. For the other measurements, the overall *p*-value was 0.010 (phase lifetime) and 0.038 (modulation lifetime). The *p*-value of the TD analysis was 0.024 and thus statistically significant. The comparison of the FD measurement series and their individual lifetime components (single *p*-values) can be seen in Table 2.

On the one hand, the longer the integration time in TD, the more events are collected, resulting in more accurate lifetimes due to better statistics. On the other hand, for the camera-based lifetime measurement, the exposure time can not be extended to arbitrary values because pixel saturation counteracts the improvement in accuracy due to the limited dynamic range of 14 bit in our case. If the exposure time is too long, the saturation distorts the measurement and causes larger error bars. To circumvent the saturation effect, several camera images with shorter integration times should be accumulated to improve the accuracy more effectively.

### 2.2. Sample Dependent Parameter

Because the setup-dependent parameters have shown consistent results for TD and FD, the parameters regarding the fluorophore itself and its microenvironment were investigated with the following setup parameters fixed: FD excitation of 405 nm and 40 MHz, while for TD, the excitation wavelength was set to 780 nm (2P excitation at 390 nm) and 80 MHz. The values for the exposure or integration time are set individually depending on the fluorescence intensity. For all coming figures, only the phase lifetime is presented because up to this point, the consistency of phase- and demodulation-lifetimes is convincingly demonstrated. In addition to the *p*-values for the fluorophore-dependent parameters, *t*-tests were carried out to check if linear correlations are statistically significant [17].

#### 2.2.1. Concentration

Although fluorescence is an intra-molecule relaxation process, some research groups were able to experimentally demonstrate a dependency on the molecular concentration, which suggests an inter-molecular influence. From moderate to high concentrations, an increase in the lifetime [12] was noticeable, while for even higher concentrations, drastic lifetime reductions occurred [6]. Inter-molecular interactions may cause these effects, which are, for example, re-absorption, self-quenching or other energy transfer mechanisms. The spectral overlap between the absorption and emission spectrum determines how strongly these mechanisms have an impact on the measured lifetime.

For the investigation of these effects, two dyes were chosen, which were measured in a concentration range starting from $10^{-1}$ M down to $10^{-5}$ M in the TD and the FD. The first used dye is lucifer yellow dissolved in ethanol, which has a small spectral overlap of its excitation and emission spectra. Fluorescein (dissolved in water) is used as the counterpart, which has a very large spectral overlap and should show a strong lifetime change in concentration-dependent measurements (Figure 2A).

As can be derived from Table 3, the series of measurements for lucifer yellow exhibits an approximately constant lifetime of around (10.289±0.371) ns for the FD over the concentration range of $10^{-5}$ to $10^{-4}$ M. The TD shows an increase from 8.837 to 11.476 ns presumably due to a re-absorption process. For high concentrations beginning from $10^{-3}$ M, a slight decrease in TD and FD lifetime measurements can be detected that may result from energy transfer mechanisms. For a concentration of $10^{-1}$ M, the FD-results showed an average lifetime of 9.685 ns. The small effects of re-absorption and energy transfer for lucifer yellow can be seen in its spectrum in Figure 2B. Within the limits of the measurement accuracy, an agreement of the TD with the FD method could be shown. Furthermore, the *p*-values (Table 3) imply congruous experimental data. The phasor plot illustrates the consistent lifetime, which is characterized by the point cloud along the semicircle (Figure 2).

The second investigated fluorophore is fluorescein, whose spectral overlap between the absorption and emission spectra is large compared to lucifer yellow, as can be seen in Figure 2B. The TD-derived lifetimes show an increase in concentrations of $10^{-5}$ M to $10^{-3}$ M, mainly through the re-absorption process (Table 3). If the concentration is increased further (up to $10^{-1}$ M), a strong reduction in the lifetime to a value of 0.209 ns can be detected caused by self-quenching effects [18]. Compared to FD, the rise of the lifetime for concentrations of $10^{-5}$ M to $10^{-3}$ M is larger and possible due to the different absorption processes. The self-quenching caused a reduction to a value of 0.380 ns for $10^{-1}$ M, which is illustrated in Figure 2B.

The visual representation of the lifetimes by the phasor plot illustrates the self-quenching effect, which is characterized by an elongated point cloud distribution along the universal circle. By comparing the phasor plot of the TD and the FD, a slightly different distribution can be observed, which is also obvious by looking at the *p*-values in Table 3.

#### 2.2.2. Solvent Polarity

The fluorescence lifetime of a fluorophore changes based on the surrounding medium due to induced conformational changes [19]. For a variety of applications, these media must be chosen accordingly.

The extent of the solvent influence was measured by using both FLIM setups on rose bengal. Figure 3A visualizes the results for the fluorescence lifetimes in relation to the used media. The main influencing factor of the solvent is its polarity, therefore, a direct correlation can be seen (Figure 3B). An increase in the polarity results in a reduction in the lifetime and vice versa.

The graphical presentation (Figure 3C) of the lifetimes measured by TD as well as FD shows consistent results within the standard deviation. The lifetimes with the respective *p*-values for each solvent are given in Table 4.

The phasor plot shows the individual point clouds of the different measurements for the corresponding media. Due to the linear relationship between the lifetime and the polarity change in the solvent, it can be stated that the polarity increases along the semicircle.

#### 2.2.3. Temperature

Although there are well-known temperature influences on the fluorescence lifetime, which are treated later in the discussion section, this investigation uses fluorescein as an example to show the opposite. Measuring fluorescein ($10^{-3}$ M in ethanol), the lifetime curve for different temperatures in the range of 21–42 ${}^{\circ }$C is shown in Figure 4A and all individual lifetimes in Table 5. Using linear regression, a slope of 0.0021 ns/${}^{\circ }$C in the TD and a slope of 0.0015 ns/${}^{\circ }$C for the FD technique can be obtained. For most FLIM setups, accuracy in the two- to three-digit picosecond range is possible. The fluorescence lifetime change resulting from the temperature variation for fluorescein can thus be neglected. Furthermore, the standard deviation of each measurement showed no strong variation.

A *t*-test was used to check the consistency of the slopes determined and their intercept for TD and FD. A significance value of $T_{Slope}=0.00313$ was determined for the slope and $T_{Axis}=0.00689$ for the axis intercept. Both values are thus below the significance level of 0.05 and are consistent.

The phasor plot analysis illustrates the mono-exponential behavior of the temperature dependency measurement due to the point cloud on the semicircle (Figure 4A). Since the values change only by a few picoseconds per measurement, all measurements together result in a uniform distribution.

#### 2.2.4. pH-Value

The pH-value of a fluorophore solution is of central importance during a fluorescence lifetime measurement. Based on the chemical composition of the dye, different spectral property changes can be obtained related to the pH-value. Therefore, a pH series of a $10^{-4}$ M concentration of fluorescein in water was measured in the TD and FD. The adjusted pH-values cover a range from 3 to 12. The detected lifetimes show a plateau for strongly acidic and alkaline pH-ranges, whereas the transition between them shows an approximately linear dependence. The resulting sigmoid function showed an overall lifetime change of ca. 1.5 ns for the adjusted pH range (Figure 4B and Table 6). The linear section from pH 6 to 9 was fitted and a *t*-test was performed. The slope of the TD was (0.336±0.022) ns and for the FD (0.348±0.028) ns. The *t*-test for the slope has a value of $T_{Slope}=0.039$ and for the axis intercept $T_{Axis}=0.021$.

Furthermore, for the pH dependence, comparable results within the respective standard deviations could be retrieved for TD and FD. With the aid of the phasor plot, the lifetime change can be easily tracked and, with the appropriate calibration, also mapped to the corresponding pH level of the microenvironment (Figure 4B).

#### 2.2.5. Quencher

Similar to the fluorescence lifetime dependency on the microenvironment (solvent polarity, temperature, pH), there are also other possibilities to intentionally quench a dye and thus obtain a shortening of the lifetime. For example, this can be utilized to mimic pathologically relevant coenzymes such as nicotinamide adenine dinucleotide (NADH) or flavin adenine dinucleotide (FAD) in optical phantoms used for clinical studies [20]. Here, coumarin 1 and coumarin 6 have similar spectra to the coenzymes but different lifetimes. To fix this issue and to fine-tune the desired lifetime of the phantom, quenching materials such as 4-hydroxy TEMPO are applied.

For the demonstration of the artificial quenching effect, this system is used as well. The starting solution was a coumarin 1 and a coumarin 6 mixture in ethanol with a concentration of $2\;\cdot \;10^{-4}$ M. The initial measurements showed a consistent lifetime for coumarin 1 of $\tau_{TD}=(3.239\pm 0.017$) ns in TD and $\tau_{FD}=(3.268\pm 0.097$) ns in FD (Figure 5A), and for coumarin 6 of $\tau_{TD}=(2.663\pm 0.013$) ns and $\tau_{FD}=(2.615\pm 0.118$) ns, respectively (Figure 5B). Shortened lifetimes could be produced by adding different concentrations of 4-hydroxy TEMPO, which are shown in Figure 5.

The phasor plot illustrates the shortening of the lifetime. The obtained values together with the *p*-values are listed in Table 7.

## 3. Discussion

By comparing TD- with FD-FLIM data of various fluorophores in different microenvironmental conditions and measured with different excitation and detection schemes, a statistical proof could be given that both techniques deliver equal lifetime values and become visible when using the phasor plot evaluation.

### 3.1. Setup Dependent Parameter

Specifically, experiments were performed by changing system parameters such as one- and two-photon excitation at various wavelengths, detection of many photons with a camera or single photons with a point detector, and using different modulation frequencies and exposure times (FD-FLIM) or repetition rates and acquisition times (TD-FLIM), respectively. All results retrieved with either technique were consistent, as demonstrated by statistical analysis. This remarkable outcome is based on the FLIM-advantage, which is extensively discussed in Section 4.1 of this manuscript.

#### 3.1.1. Wavelength

The results showed that the lifetime data are independent with respect to the wavelength and to the kind of absorption process (1P- or 2P-excitation). The Einstein model for a two-level system gives an accurate explanation for both observations. Here, the fluorescence intensity resembles the population in the excited state, whereas the lifetime corresponds to the inverse radiative transition rate described by the so-called Einstein coefficient [21]. In this picture, the absorption process leading to the population in the excited state depends not only on the transition rate but also on the intensity of the excitation light source and how well it matches the photo-physical properties of the molecule. This is worth considering because, for excitation with two photons, one would expect to double the wavelength of the laser to reach the same excitation state as with one photon. However, it was demonstrated already in 1995 by Watt W. Webb and Chris Xu that 2P absorption spectra are most likely blue shifted to the 1P spectra because the cross sections of excited singlet states are not the same for 1P and 2P excitation [22]. This leads to different excitation probabilities for the various wavelengths and, thus, different fluorescence intensities. Interestingly, the radiative transition to the ground state seems to always start from the same excited singlet state. Therefore, the transition rate responsible for the lifetime is not affected by the kind of excitation process giving rise to the FLIM-advantage (see Section 4.1).

#### 3.1.2. Modulation Frequency

For both techniques, the modulation frequency defines the range in which the fluorescence lifetime can be measured. While the FD laser can be modulated internally, the TD laser is limited by its repetition rate. Therefore, only the FD-FLIM setup was tested for different modulation frequencies showing a consistent lifetime in the range of 10–40 MHz. While the modulation frequency decreases, a slightly larger standard deviation can be noticed that results from the tangent function in Equation (Equation 10).

#### 3.1.3. Integration Time

For different integration times in both techniques, an overall consistent lifetime for rose bengal in ethanol could be achieved (Figure 1D and Table 2). Whereas the TD-FLIM setup benefits from a longer integration time and statistics, the FD-FLIM setup suffers from overexposure and pixel saturation. This context can also be seen in the standard deviations listed in Table 2.

### 3.2. Sample Dependent Parameter

More interestingly are the sample-dependent parameters because they are responsible for differing lifetimes if comparing experiments performed under circumstances that are not always openly reported. One important goal of this extensive investigation was to demonstrate as many influencing parameters as possible to raise awareness among FLIM microscopists for taking full control over their experiment.

#### 3.2.1. Concentration

The fact that the fluorescence lifetime is subject to a statistically independent process leads to the assumption that the concentration of a dye has no influence on this parameter. Even the fluorescence of only a small number of fluorophores is sufficient to record a decay curve if the integration/exposure time is set accordingly long.

Here, several effects cause variations in lifetime, as can be seen in Table 3 and Figure 2. For example, the apparently longer lifetimes can be explained by the effect of re-absorption: one molecule relaxes back to the ground state emitting a photon that is subsequently re-absorbed by another molecule lifting it to the excited state [23]. In contrast to the true lifetime τ, which is not affected, the detected lifetime $\tau_{Det}$ is prolonged because many photons may reach the detector only after this secondary step. Parameters that influence this process are the quantum yield Φ, as well as the concentration-dependent spectral overlap *J*(λ,*C*) of the fluorophore (Equation (Equation 1)).
(1)$$\tau_{Det}=\frac{\tau }{1-J(\lambda ,C)\;\cdot \;\Phi }$$

The difference in the 1P- to 2P-absorption spectra, absorption cross section and, further, the quantum yield suggest different $\tau_{Det}$ for 1P- and 2P-absorptions. This translates in this experiment to FD for 1P- and to TD for 2P-absorption.

Further, energy transfer mechanisms such as the self-quench effect or the Förster Resonance Energy Transfer (FRET) are responsible for the lifetime decrease in highly concentrated samples [6,24]. Here, a more efficient energy transfer occurs, where, for example, an excited molecule partially transfers its energy to a molecule in the ground state. This leads to the fact that only very fast transitions can undergo the radiative relaxation path of emission, which leads to a shortened detectable lifetime [25]. Again, the probability of these effects is based on several spectral and geometrical properties of the sample. The most important factor is the spectral overlap J(λ,C) between the absorption E(λ) and the emission F(λ) spectrum (Equation (Equation 2)). This context determines whether a concentration-dependent lifetime change can be detected experimentally or not. The higher the overlap, the larger the possible self-quench effect.
(2)$$J\left(\lambda ,C\right)=\int E\left(\lambda \right)\;\cdot \;F\left(\lambda \right)\;\cdot \;\lambda^{4}\;d\lambda$$

Fortunately, consistent lifetimes are most likely for the widely used working dilutions of around $10^{-6}$ M. However, for time-resolved measurements with high or changing concentrations, these effects have to be kept in mind.

#### 3.2.2. Solvent Polarity

In Section 2.2.2, the fluorescent lifetimes of rose bengal were related to the solvent used. The graphic representation in Figure 3A showed that the increasing lifetime could be correlated to a decrease in the solvent polarity (Figure 3B). The decisive parameter for this behavior will probably be the so-called solvatochromism [26]. It describes the spectral changes of a fluorophore while interacting with solvent molecules. A decreasing solvent polarity is understood as a negative solvatochromism, which is expressed by a higher energy difference between the ground and excited state. This change causes a hypsochromic shift, thus the blue shift of the absorption spectrum. The contrary case is a bathochromic shift and, therefore, has a lower energy difference. In addition to these spectral properties, parameters such as dipole moment, dielectric constant, refractive index and others are also changed, which are related to the observed fluorescence lifetime [27,28]. Again, time-resolved microscopy can be used to determine the environment more precisely and, in this case, to make statements about the polarity of a solvent [29].

#### 3.2.3. Temperature

When changing the temperature, the non-radiative rate $k_{NR}$, also known as the quenching rate, is affected. With the help of the Stokes–Einstein–Debye relationship, the quenching rate can be expressed with a dependence on the temperature and the viscosity [6,30]. Looking at Equation (Equation 3), it is obvious that an increase in the temperature T is related to a decrease in the detected fluorescence lifetime $\tau_{Det}$:(3)$$\tau_{Det}=\frac{1}{\frac{1}{\tau }+\frac{k_{B}T}{4\pi r^{3}\eta }}$$

In Equation (Equation 3), next to the Boltzmann constant $k_{B}$, the radius r and the viscosity η are also included. How strongly the temperature affects the detected lifetime $\tau_{Det}$ depends on the structure of the fluorophore. While some of them only show a small change, such as fluorescein in this case, other specially designed dyes show a strong and well-defined lifetime change that is used as temperature sensors [31,32].

#### 3.2.4. pH-Value

The pH-induced fluorescence lifetime change was also demonstrated by the group of Seungrag Lee, showing its importance in cancer research in modern days [33]. A chemical restructuring due to the additional uptake of H${}^{+}$ ions with increasing pH-value results in altered photo-physical properties. In addition to this, there are also specially designed dyes that use the FRET mechanism to detect pH level changes by time-resolved methods [34].

#### 3.2.5. Quencher

With quenching reagents such as 4-hydroxy TEMPO, it is possible to actively shorten the fluorescence lifetime due to a quenching process. The two most common quenching mechanisms are the following: The first one is the so-called static quenching process. In this situation, the quenching molecule, together with the fluorophore, creates a non-fluorescent complex. The outcome is a reduced concentration of fluorescent molecules leading to a weaker intensity. The radiative transition rate remains the same, and therefore, no fluorescence lifetime change can be noticed [35]. The second mechanism is the dynamic quenching effect. In this situation, a collision between the quenching molecule and the excited fluorophore takes place. Due to this collision, the fluorophore converts to the ground state without emitting a photon and therefore shortening the fluorescence lifetime [35]. The concentration of the quencher determines the extent to which the lifetime is shortened. This makes it possible to set almost any time constant desired. For different tasks, it is therefore possible to mimic defined biological components, for example, NADH and FAD by coumarine 1 and coumarine 6 [20]. With the help of the Stern-Volmer representation, it is possible to distinguish wherever a dynamic or static quenching process takes place [35]. For the sake of completeness, it has to be mentioned that next to the two above-mentioned quenching mechanisms, the FRET process [36] and proton/electron transfer reactions could possibly create a shortened lifetime as well [37].

### 3.3. Technique Comparison

Nevertheless, there are some differences between both techniques that are necessary to mention here. First, the image acquisition is either serial in TD or parallel in FD. In TD, the laser is focused and scanned over the sample. This takes more time than imaging the entire field of view at once with a camera, such as in FD. Therefore, the FD approach is faster compared to TD. On the other hand, two-photon excitation is only possible with focused light because of the high intensities needed for the nonlinear optical process to occur. The associated longer wavelengths lie in the near infrared, which is related to an optical window for biological tissue [38]. This enables deeper optical penetration depths by the excitation laser light in highly scattering media. Another advantage of two-photon over one-photon excitation is the optical sectioning capability, i.e., the sample can be virtually sliced like in confocal microscopy [7,39]. In conclusion, the TD technique plays out its potential the most in multiphoton microscopy, where an expensive ultrashort pulsed laser is required. On the other hand, if acquisition speed is of utmost importance, the FD technique clearly wins the race.

In addition to the acquisition, the data post processing and representation is complementary. While for the TD, no prior knowledge of the detected lifetime is needed, for the FD, more accurate values can be achieved if the modulation frequency of the excitation light is adjusted accordingly. Further, the extraction of the lifetime in the TD is complicated by the fact that dependent on the sample system, a mono- or multi-exponential fit has to be calculated. However, the interpretation is relatively easy, while for each exponential fit, a single lifetime and amplitude are calculated. In the FD, the raw data result in two lifetime components that are difficult to understand and illustrate in a concordant form.

With the help of the phasor plot, a graphical representation can be created that overcomes the above-mentioned problems. While the fit algorithm in the TD is omitted, the two components in the FD are compressed to a single vector. The phasor plot also helps to understand the lifetime distribution in an intuitive way. By comparing these graphs in biological contexts, changes in fluorophore ratios, quenching effects or other events can be easily observed and interpreted [40,41].

## 4. Materials and Methods

### 4.1. Theoretical Background

In order to understand the FLIM-advantage, the underlying quantum process is described briefly. The excitation photon elevates the sample molecules to an excited electronic state where they remain for a stochastic time period before they return to the electronic ground state either by radiative relaxation ($k_{R}$) under emission of one photon per molecule or by non-radiative relaxation ($k_{NR}$) through one or more so-called dark channels. The overall decay rate k is given by Equation (Equation 4):(4)$$k=k_{R}+k_{NR}$$

Using this definition, the decay of the population *N* of the excited state is governed by the following differential equation:(5)$$\frac{dN}{dt}=-k\;\cdot \;N$$

The solution to this is a single-exponential function:(6)$$N\left(t\right)=N_{0}\;\cdot \;\exp\left(-k\;\cdot \;t\right)$$

The initial population $N_{0}$ is dependent on the excitation and, therefore, susceptible to the kind of excitation (1P or 2P) and the efficiency thereof, which is, besides other influences, defined by the excitation wavelength. The signal intensity corresponds to the number of emitted photons and is thus the negative rate described in Equation (Equation 5), that is k·N(t). The product reflects not only the decay rate *k* but also the amount of excited molecules *N*—the better the excitation, the stronger the signal. However, if bleaching occurs, the initial population $N_{0}$ diminishes, and the signal drops accordingly.

On the other hand, FLIM only determines the decay rate *k* defined in Equation (Equation 4) and does not share this flaw of intensity measurements. However, it should be noted here that, though FLIM detects photons associated with the radiative channel $k_{R}$, it also reflects the non-radiative channels $k_{NR}$ and, therefore, the overall rate *k*. Therefore, one must distinguish the detected lifetime $\tau_{Det}=\frac{1}{k}$ from the so-called true lifetime $\tau =\frac{1}{k_{R}}$. This is the reason for the differing lifetimes reported in the literature: experiments performed under different circumstances incorporate different dark channels that lead to the above-mentioned discrepancies in $\tau_{Det}$. What makes it even more complicated is the fact that for a variety of experimental scenarios, Equation (Equation 6) may be written as a series of exponential functions accounting for several fluorescent components present in the sample. For convenience, this series shall be written in terms of the signal intensity I(t) instead of the population N(t):(7)$$I\left(t\right)=\sum_{n=1}^{N}I_{n}\;\cdot \;\exp\left(-\frac{t}{\tau_{n}}\right)$$

#### 4.1.1. Time-Domain (TD)

In TD-FLIM, a time-correlated single photon counting (TCSPC) device measures the stochastic time period between excitation and emission of the fluorescent molecule, which determines the arrival time of each measured photon. Although advanced versions such as rapidFLIM [42] increased the detection rate considerably, the latter is still limited because the detection of single photons is a prerequisite of this technique. For data analysis, sufficient events are required for making the subsequent histogram of the arrival times statistically significant. Furthermore, the origin of each photon is localized by scanning the sample point-wise in a serial fashion. Therefore, statistics based on single-photon detection combined with serial point-scanning of the sample make the acquisition time long compared to frequency domain FLIM.

In this experiment, all samples were made with dye solutions containing neither structures nor different components to increase the validity of the results. Therefore, the measured histograms were fitted with a single-exponential reconvolution fit (Equation (Equation 8)), where the instrument response function (IRF) is convolved (⊗) with the intensity decay curve including a background B.
(8)$$FIT\left(t\right)=IRF⊗I_{0}\;\cdot \;\exp\left(-\frac{t}{\tau }\right)+B$$

The samples were scanned over a range of 200 times 200 pixels to average over the many molecules of the solutions. As expected, the recorded lifetimes of all pixels showed a statistically normal distribution. For calculating the expectation value and the standard deviation of the measurements, a Gaussian fit function was used.

#### 4.1.2. Frequency Domain (FD)

In FD-FLIM, consecutive widefield images are taken in a rapid sequence with a camera to extract the lifetime for each pixel in parallel. Using specially developed sensors, these sequences can be recorded very fast and efficiently in a homodyne or heterodyne fashion [11]. Compared to conventional image sensors, each pixel of this sensor has two charge collection sites, including a switch that enables fast recording of two consecutive images. The switch is synchronized to the modulation of the excitation laser so that each image corresponds to half a period of the modulation. Please see [43,44] for a detailed description.

The excitation *E*(*t*) of the sample is performed using a sinusoidal modulated light source and can be expressed as follows:(9)$$E\left(t\right)=A_{Exc}+B_{Exc}\;\cdot \;\sin\left(2\pi \;\cdot \;f_{Mod}\;\cdot \;t+\phi_{Exc}\right)$$

The parameter $A_{Exc}$ describes the offset, $B_{Exc}$ the amplitude and $\phi_{Exc}$ the phase of the excitation. With previous knowledge of the lifetime range τ to be measured, one can choose the optimal modulation frequency $f_{Mod}$ in the experiment according to Equation (Equation 10):(10)$$f_{Mod}=\frac{1}{2\pi \;\cdot \;\tau }$$

In analogy to a forced oscillation, the generated fluorescence *F*(*t*) is delayed, as well as demodulated (Equation (Equation 11)):(11)$$F\left(t\right)=C_{Em}+D_{Em}\;\cdot \;\sin\left(2\pi \;\cdot \;f_{Mod}\;\cdot \;t+\phi_{Em}\right)$$

The modulation depths *M* of the excitation E(t), as well as of the fluorescence emission F(t), are defined in Equations (Equation 12) and (Equation 13):(12)$$M_{Exc}=\frac{B_{Exc}}{A_{Exc}}$$
(13)$$M_{Em}=\frac{D_{Em}}{C_{Em}}$$

The transfer function between excitation E(t) and fluorescence F(t) shows a demodulation (Equation (Equation 14)) and a phase shift (Equation (Equation 15)), which are directly related to the photo-physical properties of the sample.
(14)$$M=\frac{M_{Em}}{M_{Exc}}$$
(15)$$\phi =\phi_{Em}-\phi_{Exc}$$

Finally, with definitions made in Equations (Equation 14) and (Equation 15), two lifetimes $\tau_{M}$ and $\tau_{P}$ can be derived in the FD approach:(16)$$\tau_{M}=\frac{\sqrt{1-M^{2}}}{2\pi \;\cdot \;f_{Mod}\;\cdot \;M}$$
(17)$$\tau_{P}=\frac{\tan(\phi )}{2\pi \;\cdot \;f_{Mod}}$$

For single-exponential systems, both lifetimes $\tau_{M}$ and $\tau_{P}$ are the same. However, if the sample contains several fluorescent components with lifetimes lying in the range between $\tau_{1}$ and $\tau_{2}$, the demodulation lifetime $\tau_{M}$ differs from the phase lifetime $\tau_{P}$ [45]. Here, no direct assignment to the individual lifetimes of the components can be made. In this case, the single lifetime τ in Equation (Equation 10) should be substituted with the expression $\sqrt{\tau_{1}\;\cdot \;\tau_{2}}$ for choosing the best possible modulation frequency when performing the experiment.

The FD camera (pco.FLIM, pco AG, Germany) used in the experiments detects the two components $\tau_{M}$ and $\tau_{P}$ with a homodyne detection scheme [43,46]. In contrast to TD-FLIM, an intensity-modulated diode laser was used for widefield illumination that excited the samples via one-photon absorption. Analogous to the TD evaluation, all detected pixel lifetimes are collected in a histogram and evaluated using a normal distribution fit (Figure 6B).

#### 4.1.3. Phasor Plot Approach

Assuming that the emitted fluorescence of a fluorophore has a linear time-invariant behavior, it is possible to measure the lifetime in the TD as well as in the FD [46] without loss of information. For both approaches, however, the lifetime is not measured directly but has to be extracted from the data by subsequent analysis. Especially in the TD, this often leads to erroneous results depending on the selected fit function. Using the FD technique instead, the statistical fit is omitted, but the fact that two lifetimes are calculated complicates the interpretation of the results if the sample contains not just one but several components.

The strength of the phasor plot lies in the consistent representation of lifetimes measured with either technique. Furthermore, the fit algorithm needed in TD becomes obsolete, and the ambiguity present in FD is avoided [47,48]. For this expedient representation, the respective pixel information is transformed into a vector with a defined magnitude and angle. Because the pixel information differs in FD- and TD-FLIM, the two vector components *G* and *S* are calculated in different ways. In the FD, the transformation is based on the demodulation $M_{x,y}$ and the phase shift $\phi_{x,y}$ for each pixel (x,y):(18)$$G_{FD}\left(x,y\right)=M_{x,y}\;\cdot \;\cos\left(\phi_{x,y}\right)$$
(19)$$S_{FD}\left(x,y\right)=M_{x,y}\;\cdot \;\sin\left(\phi_{x,y}\right)$$

For data collected in the TD, the decay curve I(t) for each pixel (x,y) must be Fourier-transformed and normalized, as described by Equations (Equation 20) and (Equation 21):(20)$$G_{TD}\left(x,y\right)=\frac{\int_{0}^{T}I_{x,y}\left(t\right)\;\cdot \;\cos\left(2\pi \;\cdot \;f_{Rep}\;\cdot \;t\right)\;\cdot \;dt}{\int_{0}^{T}I_{x,y}\left(t\right)\;\cdot \;dt}$$
(21)$$S_{TD}\left(x,y\right)=\frac{\int_{0}^{T}I_{x,y}\left(t\right)\;\cdot \;\sin\left(2\pi \;\cdot \;f_{Rep}\;\cdot \;t\right)\;\cdot \;dt}{\int_{0}^{T}I_{x,y}\left(t\right)\;\cdot \;dt}$$

The tips of the vectors, when plotted, are located within a semicircle, also called the universal circle. This graphical representation allows a quick and detailed overview of the lifetime distribution. If the vector point ends on the universal circle, it represents a single-exponential lifetime. However, if the vector points are inside the circle, this indicates that multiple fluorescent components have been detected (Figure 6C). Using this plot, it is possible to graphically represent the correspondence of TD to FD data. When plotting the vectors of all pixels in a single phasor plot, one can check the accuracy of the measurement by comparing the sizes and densities of the respective clouds. With this representation, further analysis methods such as image segmentation can be used, where only a defined portion of the phasor plot is reproduced in the intensity image [49]. This eliminates the need for shape-based segmentations for some applications and could be a way to label images for artificial-intelligence-based algorithms [50]. Furthermore, an innovative approach to metabolic imaging could be achieved with the help of the phasor plot [51]. For example, in the field of bacteria research, different metabolic states induced by their environment could be differentiated by applying segmentation to the phasor plot representation [52].

For comparing the obtained results of lifetime data and their different influence factors, a self-made Matlab (R2020a, MathWorks, USA) program calculates the phasor plot from the raw data detected either by the TCSPC device or the FD camera. The phasor plot depiction is then represented according to an additive color overlay. For the FD- and TD-results, a separate color map was created, which generates a corresponding blended color when superimposed (here pink to white).

### 4.2. Experimental Setup

In order to obtain fluorescence lifetime data of different fluorophores, a custom-built two-photon excited fluorescence microscope (TPEFM) was used. The optical setup is built around a Nikon Eclipse Ti2 body, which was modified with various 3D-printed parts to make it suitable for a broad variety of multiphoton microscopy techniques as well as for the combination of TD- and FD-FLIM (Figure 7).

#### 4.2.1. Time-Domain Experiment

**Optical Setup:** For the TD measurements, a sample scanning approach was realized (Figure 7). An ultrashort pulsed fiber laser (FemtoFiber Dichro Design, TOPTICA Photonics AG, Gräfelfing, Germany)—referred to as **TD Laser**—emitting pulses of 95 fs duration at two wavelengths of $\lambda_{1}=780$ nm and $\lambda_{2}=1034$ nm was used for excitation. Both pulse trains with a repetition rate of 80 MHz have a maximum laser power of 100 mW each. Next, a Kepler-telescope consisting of two lenses **L1** ($f_{1}=20$ mm) and **L2** ($f_{2}=100$ mm) with a 5× magnification was integrated. It was used to overfill the back aperture of the objective, thus focusing the light to the diffraction limit. The beam enters the microscope body through the rear input port and is reflected by a dichroic mirror **D1** (F38-749, AHF Analysentechnik, Tübingen, Germany) towards the objective. This optical component is located in a filter cube. The water immersion objective (CFI Apo MRD77200, Nikon, Japan) has a 20× magnification and a numerical aperture of 0.95. A piezo stage **PS** (Nano-LPS, Mad City Labs, Madison, WI, USA) moves the sample **S** along a defined path while the laser focus stays fixed. The generated fluorescence is then collected through the same objective and passes the dichroic mirror. A selection mirror **SM** reflects the fluorescence light either to the **TD Detector** or the **FD Camera**. For TD measurements, the fluorescence is filtered by a bandpass filter **Em1** (F39-653, AHF Analysentechnik, Germany) at (550±25) nm before a lens **L3** (f=7.5 mm) focuses it onto a single photon detector (MPD-APD 50, Micro Photon Devices Srl, Bolzano, Italy).

**Electronic Signal Path:** A TCSPC card (Multiharp 150, PicoQuant, Berlin, Germany) was used to detect the laser reference signal, as well as the signal of the single photon detector to measure the time difference between them. Care was taken that each detected photon was assigned correctly to the corresponding excitation laser pulse, which is recorded by a photodiode. For this purpose, the photodiode signal is fed into an electronic pulse delay device (Picosecond Delayer, Micro Photon Devices Srl, Italy). This can shift the electrical signal with picosecond accuracy to compensate for time delays caused by the optical and electronic setup when recording the signals. The frame- and line-trigger required for image acquisition are generated by the control unit of the piezo stage.

**Control Software:** Two control programs were used for image acquisition. The Symphotime software (SymPhoTime64 Version: v2.6.5544, PicoQuant, Germany) manages data acquisition and raw data fitting. A self-made program (LabView Version: 17.0, National Instruments, Austin, TX, USA) is used to control the piezostage for sample scanning and generates the required line-start, line-end and frame trigger signals.

#### 4.2.2. Frequency-Domain Setup

**Optical Setup:** For lifetime detection in the FD, a widefield approach is required (Figure 7). Thus, the moving piezostage **PS** with which the sample is moved in the TD was omitted. Two separate laser diodes (LuxX Series, Omicron, Germany) were used, offering an internal modulation capability to generate sinusoidal radiation at wavelengths of 405 and 445 nm with 100 mW laser power for each **FD Laser**. A liquid light guide directs the light to a widefield coupling located at the rear of the microscope containing appropriate lenses to collimate (**WF1** ) and to focus (**WF2**) the emerging light into the back focal plane of the objective. A clean-up filter **Ex1** (405 nm: F49-406; 445 nm: F39-448, AHF Analysentechnik, Germany) in the beam path blocks all other wavelength components but the laser wavelength itself. To image the sample, the TD Objective **O** was reused. The laser light was separated by a dichroic mirror **D2** (405 nm: F48-425; 445 nm: F38-470, AHF Analysentechnik, Germany). For the elimination of any laser radiation or ambient light, an appropriate fluorescence filter **Em2** (405 nm: F47-424; 445 nm: F76-460, AHF Analysentechnik, Germany) was used. The selection mirror **SM** was aligned in order to focus the fluorescence signal via a lens **L4** onto an FD FLIM camera (pco.FLIM, pco AG, Kelheim, Germany).

**Signal Path:** The signal path includes two BNC cables between the camera and the laser. These cables were used to synchronize the homodyne detection method within the camera and the modulation signal of the laser. The data transfer between the control computer and the pco.FLIM camera was done via a USB 3.0 interface.

**Control Software:** The NIS Elements software (V5.0, Nikon, Japan) serves as control software. In addition to all control parameters for the lifetime detection, it is also used for analyzing the raw data. For calibration, a reference measurement with a sample of known lifetime (Starna Green, Starna Scientific Limited, Ilford, UK) has to be performed at the start of each measurement campaign.

### 4.3. Sample Preparation

To cover a wide range of dyes, a sample selection was made on the basis of spectral data to be measurable both with the TD and the FD systems. Among common fluorophores, those were selected with regard to their chemical properties to react, for example, strongly to a pH change such as fluorescein [53]. All sample fluorophores are listed in Table 8. For microscopic examination, a volume of 30 μL was placed in a sterile ibidi μ-Slide VI0.4 (80606, ibidi GmbH, Gräfelfing, Germany) and subsequently measured.

**Concentration:** All sample fluorophores here were in salt form. Using the substance-specific property of the molar weight *M* and the desired amount of salt for a specific concentration *c*, the final volume *V* was calculated according to Equation (Equation 22). For concentration series, a stock solution of V = 1.5 μL with a concentration of c=0.1 mol/L was produced that was afterward diluted to the desired concentration value.
(22)$$\;m_{Fluorophore}=V\;\cdot \;c\;\cdot \;M$$

**Solvent Polarity**: To investigate whether the polarity has an influence on the fluorescence lifetime, the dyes were dissolved in different solvents (Table 9).

**Microenvironment:** As variable environmental parameters, the pH level and the temperature of the solvent were varied. The pH level adjustments were carried out on a dye solution of 20 ml that was diluted with HCl or NaOH, respectively, to achieve pH-values in the range between 3 and 12. The exact pH-value was checked with a pH-meter (CyberScan ph 100, Eutech InstrumentsPte Lte, Singapore) measuring device before and after each lifetime measurement.

A stage top incubation system (10720, ibidi GmbH, Germany) was used for controlled temperature changes. The temperature range to be covered was between 21 and 42 ${}^{\circ }$C. For temperature verification, a chamber of the µ-Slide filled with the fluorophore solution under investigation was equipped with a temperature probe. For each measurement, a 10 min settling period was ensured to avoid possible temperature fluctuation during the measurement.

**Quencher:** Quencher experiments were made with 4-hydroxy TEMPO (176141, Sigma Aldrich Chemie GmbH, Darmstadt, Germany) in combination with coumarine 1 and coumarine 6 in two different concentrations. For coumarine 1, quencher concentration of 0.00855 and 0.234 M were used and for coumarine 6, quencher concentrations of 0.0013 and 0.713 M.

### 4.4. Statistics

To meet statistical standards, both TD and FD experiments were repeated seven times for each measurement. To confirm the reproducibility of the series, a *p*-value was calculated [54]. Therefore, a binomial distribution of τ and its error was assumed. If the *p*-value lies under a threshold of 0.05, it is statistically proven that the lifetimes that are obtained under different conditions are consistent. These conditions may be the application of a specific setup such as TD and FD or changing specific parameters such as temperature, pH level and so forth. For the graphical presentation of the measurement results, the mean values and their averaged standard deviation are displayed next to the phasor plot.

## 5. Conclusions

In this work, several experimental parameters are demonstrated to have an influence on the lifetime of specific fluorophores. This sensitivity can be used to the advantage of the experimenter: monitoring the lifetime will reveal how much a parameter changes during the measurement. On the other hand, having these findings in mind, one has to be careful to have control over all influencing parameters to make the results comparable and consistent with other measurements [55]. Therefore, it would be beneficial in future publications to also name the solvent polarity, the temperature and the pH-value next to the fluorophore. If a large spectral overlap between the absorption and emission spectrum of the fluorophore exists, its concentration should also be mentioned.

The statistical analysis of all measurements proved without exception that the results of the TD setup are consistent with the ones obtained with the FD setup. Therefore, it can be concluded that both setups produce reliable data if all influencing parameters are under the control of the experimenter. Both techniques have the advantage in common of not relying on the way the fluorophore is excited. Therefore, their independence from the excitation laser wavelength or from the number of photons absorbed to transfer the fluorophore to its excited state was demonstrated.

While the discussion section explains which fundamental process lies behind each influencing parameter, it is possible to use this information and create fluorophores that function as biological sensors. For example, fluorescein is used in clinical studies to image the pH-value and infer cancer in breast tissue [33]. Further, special dyes were developed that can monitor the intracellular viscosity through the fluorescence lifetime [56]. Other fluorophores give the possibility to detect intracellular temperature changes [31]. Each experiment can thus benefit from selecting the right dye for the particular parameter that has to be determined.

## Figures and Tables

**Figure 1 ijms-23-15885-f001:**
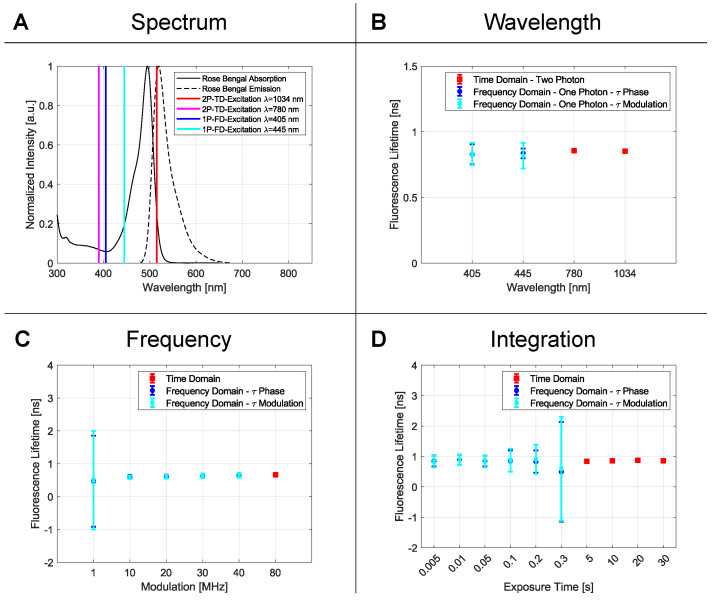
In (**A**), the absorption and emission spectrum of rose bengal and the individual excitation wavelengths are shown. The blue (405 nm) and cyan line (445 nm) represent the FD Wavelengths. For TD FLIM, the two-photon excitation wavelengths are shown in pink (two-photon excitation at 390 nm) and red (two-photon excitation at 515 nm). Although the laser lines are only slightly within the absorption spectrum, a sufficiently high fluorescence can be generated. In (**B**), the obtained fluorescence lifetimes for the wavelength dependency measurements were presented. In blue (phase lifetime) and cyan (demodulation lifetime), the results for the FD are shown, and in red the results for the TD are shown. In section (**C**), the influence of the repetition rate (TD) and the modulation frequency (FD) are shown. For the FD, the tangential relationship leads to inaccurate results for small modulation frequencies for lifetimes in the range of 1–10 ns. As the last parameter, the time constant for the different measurement methods is illustrated in (**D**). For the FD FLIM, high exposure times lead to pixel saturation and therefore creates bigger standard deviations. For the TD, the correlation is inverse. For longer integration times, a smaller error bar is detected due to the larger number of photons that form the decay statistic.

**Figure 2 ijms-23-15885-f002:**
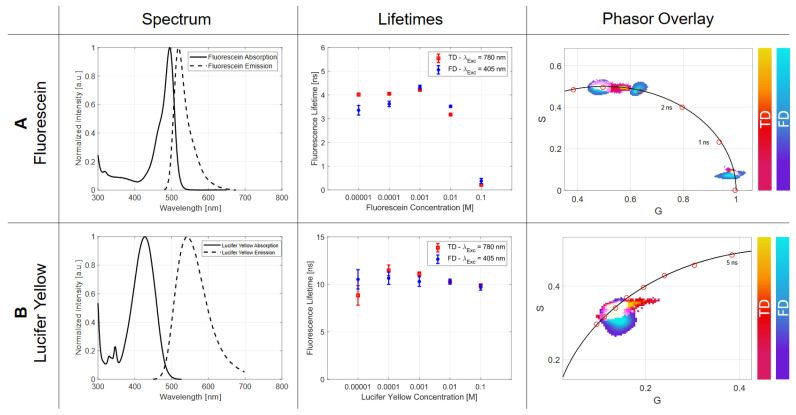
In this figure, the spectral properties of the two fluorophores used, fluorescein and lucifer yellow, are shown in (**A**) to check if the concentration could lead to an influencing parameter while measuring lifetime values, the spectral overlap must be observed. Fluorescein shows a relatively big spectral overlap compared to lucifer yellow. With the graphs in (**B**), the lifetimes of the TD are shown in red and the ones of the FD are in blue. For fluorescein, the lifetime change for high concentrations can be seen clearly, whereas the results of lucifer yellow stay nearly the same. The last column presents this relation in the phasor plot for the TD, as well as for the FD data.

**Figure 3 ijms-23-15885-f003:**
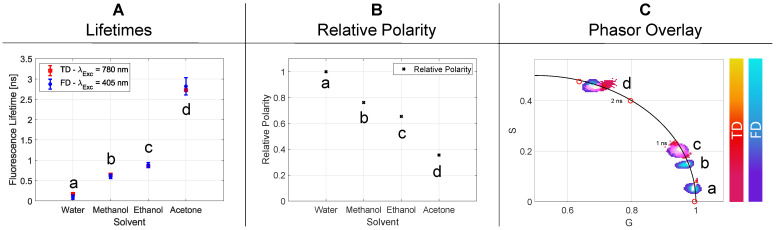
In section (**A**), the relative polarity of each individual solvent is illustrated. Compared to the obtained lifetimes in (**B**), an indirectly proportional behavior to the polarity is noticeable. Section (**C**) represents the overlayed phasor plots of the TD and FD data.

**Figure 4 ijms-23-15885-f004:**
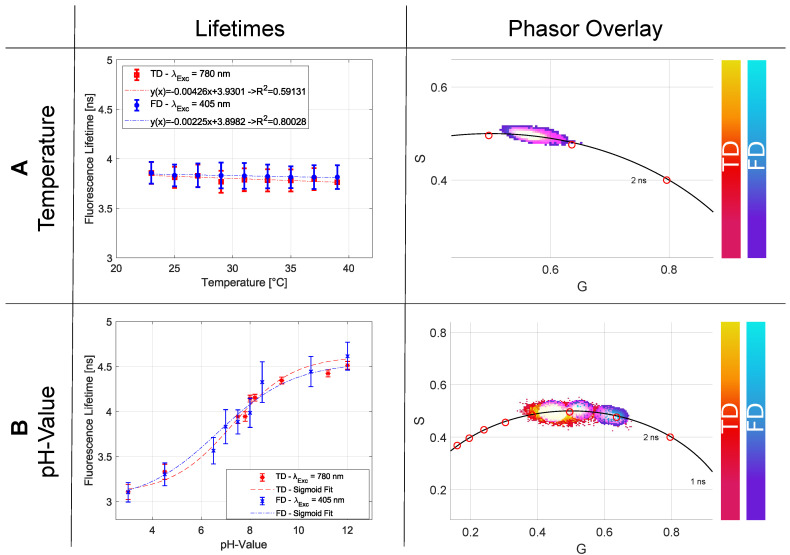
(**A**) shows the obtained results for the temperature dependency between 22–39 ${}^{\circ }$C and the slope of the fitted linear regression for fluorescein in ethanol. The overall lifetimes for the TD and FD are also represented in the phasor plot where a single point cloud indicates no significant lifetime change. In (**B**) the pH dependency for values between 3 and 12 are shown, which follow a sigmoid function.

**Figure 5 ijms-23-15885-f005:**
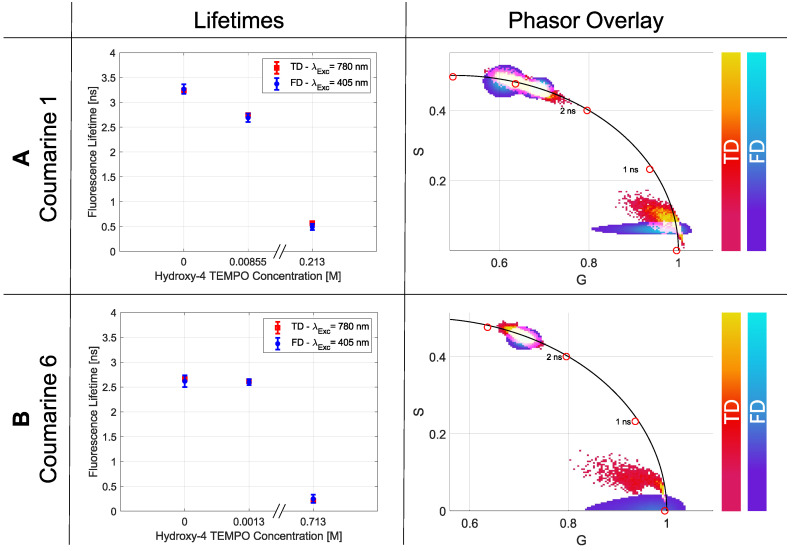
Results of the fluorescence lifetimes and phasor plot for the quenching series of coumarine 1 are illustrated in (**A**,**B**) for coumarine 6. With the help of 4-hydroxy TEMPO, it is possible to decrease the fluorescence lifetime to a specific value.

**Figure 6 ijms-23-15885-f006:**
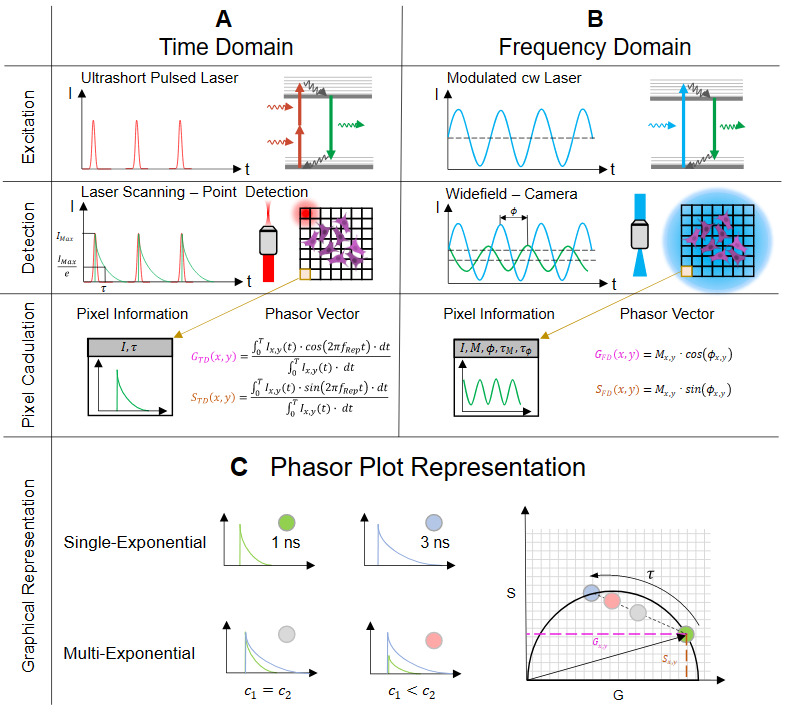
This figure shows both methods of obtaining the fluorescence lifetime in the (**A**) time-domain as well as in the frequency-domain (**B**). In the time-domain (**A**), a TCSPC device acts as a stopwatch and registers the time differences between the excitation (top row) and the arrival of the emitted fluorescence photon detected by a single photon detector (middle row). In the frequency-domain (**B**), the lifetime information is calculated by comparing the phase and amplitude between the modulated excitation (top row) and the modulated fluorescence of the sample (middle row). The obtained values in TD- and FD-FLIM (bottom row) are correlated by the Fourier transformation. (**C**) shows a way to represent both measurement results in a single plot, the so-called phasor plot. Each lifetime of a pixel is represented by a vector with the components G and S. Single exponential decays are located on the semicircle while multi-exponential decays are located within the semicircle. The position of the point cloud along the connection line between the two dyes indicates the concentration ratio c between them.

**Figure 7 ijms-23-15885-f007:**
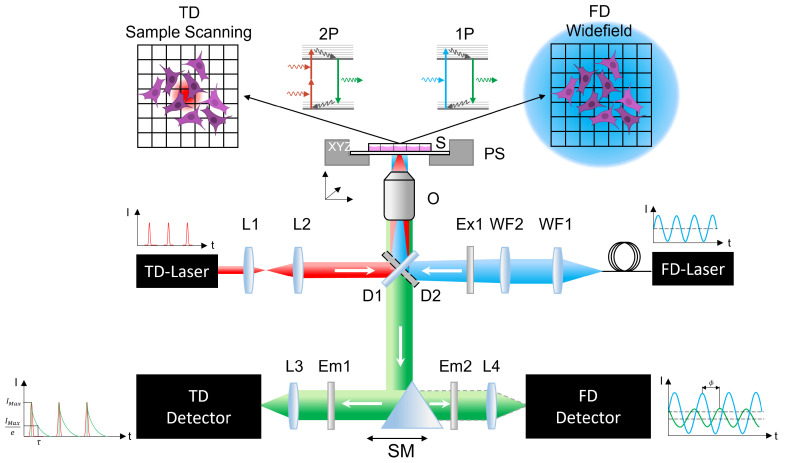
Schematic drawing of the optical setup. The excitation beam path for the TD setup is indicated in red. The fluorescence (green) for each pixel is detected with a single photon detector by switching the selection mirror **SM** to the right position. The widefield excitation beam path for the FD setup is shown in blue. Because one-photon excitation is used in FD and two-photon excitation is used in TD, the dichroic mirror **D1** differs from **D2**, thus excluding a simultaneous measurement. Instead, individual filter cubes, one containing the excitation filter **Ex1** and dichroic mirror **D1** and the other containing **D2**, are placed in the beam path depending on the measurement. L: lens, WF: widefield lens, Ex: excitation filter, D: dichroic mirror, S: sample, PS: piezo stage, Em: emission filter, SM: selection mirror, 1P: one-photon excitation, 2P: two-photon excitation, O: objective lens.

**Table 1 ijms-23-15885-t001:** Single and overall *p*-values for each frequency for the phase- and demodulation-lifetime of the FD technique, when measuring rose bengal.

Modulation Frequency						
	**1 MHz**	**10 MHz**	**20 MHz**	**30 MHz**	**40 MHz**	**Overall** * **p** * **-Value**
$\tau_{P}$ [ns]	0.468	0.599	0.613	0.628	0.645	0.048
$\sigma_{P}$ [ns]	1.387	0.056	0.053	0.04	0.052	
$\tau_{M}$ [ns]	0.501	0.589	0.602	0.631	0.645	0.046
$\sigma_{M}$ [ns]	1.499	0.055	0.055	0.072	0.071	
**Single** * **p** * **-Values**	0.063	0.034	0.021	0.027	0	

**Table 2 ijms-23-15885-t002:** Single and overall *p*-values for each exposure time for the phase- and demodulation-lifetime of the FD setup, as well as the exposure time of the TD setup for measurements on rose bengal.

Integration Time (TD)							
	**5 s**	**10 s**	**20 s**	**30 s**			**Overall** * **p** * **-Value**
τ [ns]	0.842	0.859	0.875	0.856			0.024
σ [ns]	0.020	0.015	0.013	0.011			
**Exposure Time (FD)**							
	**5 ms**	**10 ms**	**50 ms**	**100 ms**	**200 ms**	**300 ms**	**Overall** * **p** * **-Value**
$\tau_{P}$ [ns]	0.852	0.897	0.852	0.847	0.832	0.495	0.010
$\sigma_{P}$ [ns]	0.181	0.166	0.180	0.348	0.368	1.636	
$\tau_{M}$ [ns]	0.832	0.895	0.842	0.877	0.904	0.595	0.038
$\sigma_{M}$ [ns]	0.190	0.169	0.200	0.377	0.480	1.712	
**Single** * **p** * **-Values**	0.001	0.001	0.020	0.042	0.049	0.053	

**Table 3 ijms-23-15885-t003:** TD and FD fluorescence lifetimes and *p*-values obtained for the concentration series for lucifer yellow and fluorescein.

Lucifer Yellow					
**Concentration**	**0.00001 M**	**0.0001 M**	**0.001 M**	**0.01 M**	**0.1 M**
$\tau_{TD}$ [ns]	8.837	11.476	11.087	10.265	9.907
$\sigma_{TD}$ [ns]	1.027	0.565	0.221	0.053	0.049
$\tau_{FD}$ [ns]	10.532	10.646	10.294	10.288	9.685
$\sigma_{FD}$ [ns]	1.020	0.647	0.506	0.267	0.298
**Single** ***p*****-Value**	0.052	0.499	0.045	0.042	0.049
**Fluorescein**					
**Concentration**	**0.00001 M**	**0.0001 M**	**0.001 M**	**0.01 M**	**0.1 M**
$\tau_{TD}$ [ns]	4.015	4.049	4.212	3.176	0.209
$\sigma_{TD}$ [ns]	0.064	0.025	0.018	0.028	0.002
$\tau_{FD}$ [ns]	3.356	3.623	4.336	3.520	0.380
$\sigma_{FD}$ [ns]	0.203	0.115	0.075	0.045	0.106
**Single** ***p*****-Value**	0.062	0.051	0.034	0.021	0.039

**Table 4 ijms-23-15885-t004:** Calculated *p*-values, detected lifetimes and relative polarity values for different solvents.

Figure Declaration	Solvent	Relative Polarity	$\tau_{\mathrm{TD}}$ [ns]	$\sigma_{\mathrm{TD}}$ [ns]	$\tau_{\mathrm{FD}}$ [ns]	$\sigma_{\mathrm{FD}}$ [ns]	Single *p*-Value
a	Water	1	0.127	0.014	0.093	0.052	0.050
b	Methanol	0.762	0.645	0.012	0.605	0.052	0.045
c	Ethanol	0.654	0.852	0.011	0.883	0.062	0.039
d	Aceton	0.355	2.725	0.014	2.819	0.214	0.021

**Table 5 ijms-23-15885-t005:** P-values and obtained lifetimes for the temperature series of fluorescein.

Fluorescein									
**Temperature [ ${}^{\circ }$C]**	**23**	**25**	**27**	**29**	**31**	**33**	**35**	**37**	**39**
$\tau_{TD}$ [ns]	3.857	3.813	3.830	3.769	3.789	3.783	3.781	3.796	3.763
$\sigma_{TD}$ [ns]	0.110	0.107	0.113	0.112	0.115	0.112	0.111	0.111	0.0122
$\tau_{FD}$ [ns]	3.859	3.833	3.834	3.832	3.828	3.823	3.815	3.816	3.816
$\sigma_{FD}$ [ns]	0.110	0.110	0.120	0.130	0.130	0.120	0.110	0.120	0.120
**Single** ***p*****-Values**	0.016	0.038	0.018	0.041	0.038	0.038	0.034	0.021	0.040

**Table 6 ijms-23-15885-t006:** Measured lifetimes for the pH-series in the TD and FD for fluorescein.

Fluorescein									
**pH-Value**	**3**	**4.5**	**7.5**	**7.8**	**8**	**8.2**	**9.3**	**11.2**	**12**
$\tau_{TD}$ [ns]	3.107	3.330	3.944	3.945	4.144	4.154	4.345	4.424	4.514
$\sigma_{TD}$ [ns]	0.084	0.084	0.076	0.053	0.038	0.038	0.038	0.039	0.042
**pH-Value**	**3**	**4.5**	**6.5**	**7**	**7.5**	**8**	**8.5**	**10.5**	**12**
$\tau_{FD}$ [ns]	3.104	3.303	3.565	3.832	3.885	3.985	4.327	4.445	4.615
$\sigma_{FD}$ [ns]	0.108	0.127	0.146	0.189	0.134	0.162	0.226	0.168	0.155

**Table 7 ijms-23-15885-t007:** Results for the quenching experiments for coumarine 1 and coumarine 6.

	Coumarine 1			Coumarine 6		
**4-hydroxy TEMPO**	**0 M**	**0.00855 M**	**0.234 M**	**0 M**	**0.0013 M**	**0.713 M**
$\tau_{TD}$ [ns]	3.239	2.745	0.569	2.663	2.617	0.209
$\sigma_{TD}$ [ns]	0.017	0.016	0.011	0.013	0.014	0.002
$\tau_{FD}$ [ns]	3.268	2.689	0.492	2.615	2.599	0.25
$\sigma_{FD}$ [ns]	0.097	0.086	0.063	0.118	0.060	0.082
**Single** ***p*****-Value**	0.031	0.041	0.050	0.022	0.044	0.061

**Table 8 ijms-23-15885-t008:** Sample fluorophores with the respective concentration.

Number	Fluorophore	Distributor	Article No.	Molar Weight (g/mol)	Concentration (M)
1	Rose Bengal	Sigma Aldrich	330000	1017.64	10${}^{-1}$, 10${}^{-2}$, 10${}^{-3}$, 10${}^{-4}$
2	Lucifer Yellow	Sigma Aldrich	LO144	521.57	10${}^{-1}$, 10${}^{-2}$, 10${}^{-3}$, 10${}^{-4}$, 8·10${}^{-5}$
3	Fluorescein	Sigma Aldrich	46955	332.31	10${}^{-1}$, 10${}^{-2}$, 10${}^{-3}$, 10${}^{-4}$
4	Coumarin 1	Sigma Aldrich	D87759	231.29	10${}^{-1}$, 10${}^{-2}$, 10${}^{-3}$, 10${}^{-4}$
5	Coumarin 6	Sigma Aldrich	546283	350.43	10${}^{-1}$, 10${}^{-2}$, 10${}^{-3}$, 10${}^{-4}$

**Table 9 ijms-23-15885-t009:** Solvent used for the different types of fluorophores.

Solvent	Distributor	Article No.	Fluorophores
Water	Merck	1.15333	1, 2, 3
Methanol	Merck	1.06002	1, 2, 3, 4, 5
Ethanol	Merck	1.0098	1, 2
Aceton	Sigma Aldrich	270725-1L	1, 2

## Data Availability

Not applicable.

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
