# Peer review of "Comprehensive Investigation of Parameters Influencing Fluorescence Lifetime Imaging Microscopy in Frequency- and Time-Domain Illustrated by Phasor Plot Analysis"

_ijms, 2022, doi:10.3390/ijms232415885_

Round 1

Reviewer 1 Report

The authors have systematically studied how external factors during fluorescence life time (FLIM) imaging experiments affect the results obtained by using either the time vs the frequency domain versions of FLIM.

To do that, the authors have implemented the time and the frequency domain  FLIM in the same microscopy system. the y show advantages and disadvantages of the tow FLIM domains. Also, by using the phasor-plot, the  authors show consistent results between the two techniques. Also, based on their conclusions, the authors provide a quantitative guidance for selecting the most suitable technique for getting consistent fluorescence lifetimes.

The paper is well written, clearly explained and conclusions are strongly supported by their measurements and analysis. With no doubt, this paper will be of interest to the imaging community. Therefore, I suggest publication of the manuscript.

I have the following MINOR comments to the authors:

1.- By reading the introduction, it is not clear to me if any measured lifetime is different because of external conditions or because they are measured with a different domain (i.e time vs frequency)?

2.- In lines 36 to 38 the authors state that FLIM provides different results, indicating that in the literature it is possible to find such inconsistencies. It will be interesting that the authors could explicitly provide a list of references and cases in which such apparent discrepancy has been reported. Importantly, this list should include the relevant external factors around the measurements. Finally, given the apparent different results, the authors should explain what is the implication of measured lifetime in each of the reported cases. In other words, are any of these measurements leading to incorrect conclusions?  

(Same applies to lines 70 to 76).

3.- lines 44-46. Authors could explicitly mention the "certain parameters". 

4.- Lines 104-105 Authors should specify what do they mean for specialised camera and for specially developed sensors.

5.- Lines 168-169. Please further explain and add corresponding reference to the comment: "Also an innovative approach for metabolic imaging could be achieved with the help of the phasor plot". 

6.- The authors state that the increased error bars for the FD measurements in figure 3D is due to saturation. This is further discussed in lines 369 to 376 arguing a "limited" dynamic range of 14 bits. At shorter integration times (0.1-0.2s), could the authors show evidence of such saturation in their cameras?.

Reviewer 2 Report

Fluorescence lifetime provides important information regarding fluorophores that can be used in multiple applications. There are two approaches to measure and process the lifetime data: time domain and frequency domain. Both approaches are broadly present in the scientific community, making the manuscript titled “Comprehensive Investigation of Parameters Influencing Fluorescence Lifetime Imaging Microscopy in Frequency- and Time-Domain Illustrated by Phasor Plot Analysis” of great potential interest both to spectroscopy and microscopy communities. In addition, the authors report and discuss lifetimes of fluorophores at different concentrations, temperatures, solution polarity and pH values. Particularly valuable is the fact that the two different approaches were combined and implemented in the same imaging system and microscope, making then a direct comparison between the two approaches using phasor plot. Furthermore, the manuscript includes a sufficient theoretical and experimental background, it is clearly written and includes proper description of both TD and FD FLIM experimental techniques.

Concerns:

The manuscript does not include an exemplary TSCPC curve with the correspondent fit, then there is no way to estimate the fit quality and the mono-exponentiality for TD experiment.

PCO.FLIM camera requires a calibration with sample with known a priori lifetime value. I found only brief mention of this procedure in the text (line 240). What was the laser power and the exposure time used for calibration? What is the literature value of lifetime of Starna Green used for calibration? I assume the calibration at specific experimental conditions is closely related to the “saturation effect” discussed later in the text (lines 359 and 375). This also explains the increase in FD lifetime uncertainty in the longer exposure times, as shown in Figure 3D. However, this may confuse the reader as typically longer exposure time results in a better precision in lifetime estimation. I fully agree with the explanation provided by authors, however this is something related to a particular lifetime camera used for the TD experiment and this fact shell be further explained in the text, for example in the Discussion section 4.1.3.

Minor concerns:

Line 21: Reverence to for example review article is suitable after the first sentence.

Line 169: Can the authors elaborate or provide a Ref to a “metabolic imaging”?

Line 176: Past tense seems a better fit: “The optical setup is” -> The optical setup was…

Line 300: “needed detection time often causes difficulties.” Which kind of difficulties? Please provide more details.

Caption of Figure 3: spelling typo “modulation frequencys” -> modulation frequencies

Figure 4, 5 and 6: I am confused by the legend in phasor plots – which sub-population is FD and which is FD? Please find a more clear way to present the legend in this figure.

Reviewer 3 Report

This manuscript combines time domain (TD) and frequency domain (FD) FLIM into one switchable set-up and shows that both techniques lead to the same results with phasor plot evaluation. In this manuscript, the authors also studied the possible dependencies of the measurement accuracy to different parameters, including setup parameter, excitation wavelength, modulation frequency, integration time and sample parameters.

The topic is of certain interest and novelty for publication, the method is solid the manuscript is in good shape. Only few places may need to add more details:

1.      Could the authors provide some reference to support the statement in lines 36,37?

2.      Although it may be clear to some people, could the authors specify the symbols and variables in Eq. (5)?

3.      Regarding the setup in Figure 2,  

a)     What does the arrow (white and red) on the top of the sample mean? Looks like there’s no illuminator or detector on the transilluminating side.

b)     Is Kepler-telescope represented by L1 L2? If so, which one has focal length as f1 which has as f2?

c)     What’s the focal length of L3?
